# “There Is a Cat on Our Ward”: Inpatient and Staff Member Attitudes toward and Experiences with Cats in a Psychiatric Ward

**DOI:** 10.3390/ijerph16173108

**Published:** 2019-08-27

**Authors:** Cora Wagner, Undine E. Lang, Karin Hediger

**Affiliations:** 1Clinical Psychology and Psychotherapy, Department of Psychology, University of Basel, Basel 4055, Switzerland; 2University Psychiatric Clinics (UPK), University of Basel, Basel 4002, Switzerland; 3Human and Animal Health Unit, Department of Epidemiology and Public Health, Swiss Tropical and Public Health Institute, Basel 4051, Switzerland; 4Institute for Interdisciplinary Research on Human-Animal Interaction, IEMT Switzerland, Basel 4002, Switzerland; 5REHAB Basel, Clinic for Neurorehabilitation and Paraplegiology, Basel 4055, Switzerland

**Keywords:** animal-assisted intervention, cat, ward animal, psychiatry, satisfaction, atmosphere, social interaction, relationship

## Abstract

The aim of this study was to investigate inpatient and staff member attitudes toward and experiences with ward cats, and identify possible mechanisms for how cats affect patient satisfaction in a psychiatric clinic. Thirty-three inpatients diagnosed with depression or psychosis residing on wards with and without cats and 17 staff members working on wards with cats participated in semi-structured interviews using a cross-sectional study design. Data analysis included descriptive statistics and correlations. The results showed that 17 out of 19 inpatients and all the staff members liked having a cat on their ward. Further, 12 out of 14 inpatients on wards without cats would like having a cat on their ward. Inpatient perceptions of the cat’s impact on the ward atmosphere correlated significantly with their emotional relationship with the cat (*p* = 0.015, *r* = 0.561), how often they saw the cat (*p* = 0.002, *r* = 0.676), and if they liked cats in general (*p* = 0.041, *r* = 0.486). Our results highlight the positive attitudes of inpatients and staff members toward ward cats and the potential of ward cats to enhance patient satisfaction. This influence might be mediated by factors such as the frequency of contact, the relationship between each patient and the cat, and each patient’s attitude toward cats in general.

## 1. Introduction

Patient satisfaction of inpatients with mental illness is an important factor in predicting functional and clinical outcomes [1]. It is linked with quality of life and prognosis for psychological and physical conditions [2,3], level of patient functioning [3], and adherence to treatment and therapeutic success [4]. Therefore, it is crucial to understand how to enhance patient satisfaction. 

A recent investigation found that psychiatric inpatients treated on wards with cats were more satisfied compared with patients on wards without cats [5]. Research documents that animals can positively influence atmosphere, have a soothing effect on humans, and reduce stress [6,7,8,9,10,11,12,13,14]. The presence of an animal can lead to increased social interaction in humans [15,16,17], acting as social facilitators and providing social support for humans [11,18,19]. A growing body of literature documents the effects of dogs and other animals on well-being in hospitals and nursing homes, such as reducing depression [20,21] and decreasing feelings of loneliness [22,23,24] in inpatients with dementia. However, results on mental health outcomes are mixed [25]. Research also suggests that the presence of the animals in hospitals can be beneficial for staff members [9]. 

Concrete mechanisms explaining the effect of the presence of ward cats on patient satisfaction in our previous study [5] were unclear. The study used a cross-sectional design, and did not assess whether inpatients noticed or interacted with the cats or patient perception of the cats. Although previous research suggests positive effects due to the presence of an animal, no studies examined the assumed influence of animals on patient satisfaction. 

The aim of this study was to investigate inpatient attitudes toward and experiences with ward cats and identify possible mechanisms for how cats might affect patient satisfaction. We conducted semi-structured interviews with inpatients in psychiatric treatment on wards with cats and wards without cats. We additionally studied staff member perceptions on the effects of the cats.

## 2. Materials and Methods

We interviewed all 33 inpatients with a diagnosis of depression (*N* = 20) or psychosis (*N* = 13) in four wards at the University Psychiatric Clinics (UPK) in Basel, Switzerland between July and August 2017. All the wards were acute psychiatric wards with an open-door policy; two were wards for inpatients with depression and two were wards for inpatients with psychosis, and each had one ward with and one ward without a cat. Patients were assigned to wards based on their mental health problems and were not given the choice of a ward with or without a cat. We also interviewed 17 staff members working on the two wards with cats. All the cats had lived on the wards for several years, with free choice daytime access to ward common rooms, patient rooms, or the clinic’s outdoor garden. At night, the cats remained in the ward common rooms. Inpatients were not permitted to close cats into their rooms at any time, and were advised to interact with the cats in the common rooms. Staff members were responsible for cat care, such as feeding, grooming, and veterinary checks.

We interviewed inpatients and staff members after they agreed with the informed consent. The cantonal ethics committee (Ethikkommission Nordwest und Zentralschweiz, EKNZ) approved the study protocol, which was registered at clinicaltrials.gov (Project-ID: 2017-01109).

The semi-structured interview targeted inpatient attitudes toward housing a cat on their ward, inpatient interactions with the cats, and inpatient general beliefs about the effects of a cat on atmosphere, social interactions, and their own well-being. We created similar interviews for inpatients on wards with and without cats and for staff members on wards with cats. The interview for inpatients on wards with cats contained 31 items: 11 rating questions, 13 polar questions, and 7 open questions (see Appendix A). We did not tell patients that the interview was specifically about cats, but stated that it was about their general perception of the ward. The first part of the interview covered eight items on patient sociographic and hospitalization information: how long the patient had been on the ward and if the hospitalization was at their own request, where and how they spent their free time in the clinic area, what they liked about the ward, how satisfied they were with the ward, and what aspects they perceived important for ward atmosphere. This was followed by 22 cat-related items. Two of these 22 items determined if the patient knew that a cat was living on their ward and how often they saw the cat. Four items targeted the relationship between the inpatient and the cat, i.e., the emotional relationship and how much time they spent with the cat. One item asked if the patient thought the cat was affectionate. Three questions targeted patient attitude toward housing a cat on their ward. Five items asked patients about the ward cat’s impact on them and their stay at the clinic. The last seven items assessed the inpatient’s pet history, previous experiences with pets, whether they liked animals and cats, and if they had a cat hair allergy.

The interview for inpatients on wards without cats contained 24 items: 8 rating questions, 9 polar questions, and 7 open questions. The first part of the interview was identical to the interview with inpatients on wards with cats. After that, inpatients on wards without cats were asked 14 cat-related questions. One question assessed if inpatients knew about the cats on other wards. Two items targeted the patient’s attitude about potentially housing cats on their ward. Two items asked inpatients about a cat’s potential impact on their ward, and one item asked about a cat’s potential impact on the patient. Two items assessed if inpatients would like to have a cat or other animal on their ward. Seven items investigated inpatient’s pet history, previous experiences with pets, whether they liked animals and cats, and if they had a cat hair allergy. 

The interview for staff members working on wards with cats contained 31 items: 12 rating questions, 16 polar questions, and 3 open questions. Similar to the interview conducted with inpatients, the first part of the interview assessed sociodemographic information, how long they had worked on the ward, their satisfaction with the ward, and what they perceived as important regarding ward atmosphere. After this, there were 26 cat-related items. Two items examined if the staff member knew that a cat lived on the ward and how often they saw the cat. One item asked if they thought the cat was affectionate. Three items assessed the staff member’s relationship to the cat. Three items asked the staff member about their attitude toward housing a cat on their ward. Two items assessed who was responsible for the cat, and if the cat required any additional effort by the staff member. One item asked staff members how the cat influenced their work satisfaction. Three items assessed how staff members perceived the cat’s impact on ward atmosphere. Four questions detected the staff member’s perception of the influence of the ward cat on inpatients. Seven items assessed staff member’s pet history, if they liked animals, and if they had a cat-hair allergy. 

The rating questions used a 6-stage Likert scale, with 1 indicating “no approval”, “negative”, “not at all”, or “never”; and 6 indicating “high approval”, “positive”, “very much”, or “very often”. For interpretation of results, we combined scores: 1 and 2 were considered as “no approval”, “negative” or “not at all”; 3 and 4 were considered as “medium”; and 5 and 6 were considered as “high approval”, “positive”, or “very much”. The statistical analyses were done without dichotomization. The question of ward cat impact on ward atmosphere in the interviews with inpatients on wards with cats had a reversed scale, which we recoded before data analysis. The polar questions (yes/no) explicitly examined inpatient opinions about suggested mechanisms. The interviews lasted approximately 15 minutes for each patient. 

We analyzed patient answers on the open questions with Mayring’s qualitative content analysis approach [26]. First, we transcribed the answers of each patient. Second, we analyzed all the components of each answer of each inpatient and built categories. Third, the first author and two independent raters broadly summarized the categories. Fourth, we compared the categorizations and created final categories. Lastly, we counted the frequency for each category. 

We used descriptive statistics to describe the patient answers. Spearman’s correlation coefficients determined the relationship between different questions, since data were not normally distributed. We set the confidence interval at 95%. We used SPSS, Version 25 (SPSS Inc., Chicago, IL, USA), for statistical analyses.

## 3. Results

Thirty-three inpatients (18 = female and 15 = male; mean age of 52.56 years, range 19–86) completed the interview, of whom 19 were on a ward with cats (9 inpatients with psychosis and 10 inpatients with depression) and 14 were on a ward without cats (4 inpatients with psychosis and 10 inpatients with depression). 

Seventeen staff members (11 = female and 6 = male; mean age of 40.60 years, range 21–62) participated in interviews, all of whom worked on wards with cats. 

There was no difference between ward satisfaction between inpatients treated on wards with cats (M = 4.95, SD, 1.03) compared to inpatients on wards without cats (M = 4.71, SD = 0.73; *z* = −0.91, *p* = 0.362, *d* = 0.27).

### 3.1. Inpatients Treated on Wards with Cats

The results of the open-ended questions revealed that all the inpatients listed positive aspects of housing a cat on their ward, while 6 of 19 named negative aspects after being specifically asked about them. Ten inpatients reported the contact with an animal as a positive aspect (e.g., animals are real, do not pretend, can be petted), and 9 of 19 inpatients said that cats had positive effects on their state of mind (e.g., cat has a calming effect, increases happiness, reduces stress). Thirteen of 19 inpatients did not report any negative aspects when asked about these. The negative aspects of a cat included potential but not actual problems such as hygiene, allergies, cat welfare, and if someone did not like cats or was afraid of them. 

The rating questions revealed that 17 out of 19 inpatients reported positive feelings about the cat on their ward, whereas two reported neutral feelings, and no one reported negative feelings about the cat (see Table 1). Seven of 19 inpatients believed the cat had a positive effect on the ward atmosphere. Six inpatients stated that the ward cat positively influenced their stay at the clinic. Nine out of 19 inpatients thought the ward cat positively influenced their emotional well-being, 10 believed the cat had a medium effect, and no inpatient thought the cat had a negative effect on their emotional well-being. Ten out of 19 inpatients reported having no close emotional relationship with the cat, while only 3 inpatients indicated a close emotional relationship. Further, 13 out of 19 inpatients reported spending up to 10 minutes with the cat daily, 1 patient spent up to 60 minutes daily with the cat, and 5 reported spending no time with the cat. The results from the polar questions showed that all the inpatients noticed the cats and knew them (see Table 2). Twelve out of 19 inpatients did not believe that the ward cat helped them feel more comfortable in the clinic, and 11 did not believe that the cats led to conversations with other inpatients. Six out of 19 inpatients agreed that the ward cat was affectionate and contact with the cat was possible, but only 4 actively sought such contact. Moreover, 17 disliked the idea of not having a cat on their ward in the future, and 15 did not prefer another ward animal. Three of the four inpatients preferring another ward animal suggested a dog, while one abstained from a decision.

Inpatient perception of the ward cat’s impact on ward atmosphere was significantly correlated with their emotional relationship with the cat (*r* = 0.561, *p* = 0.015) and their attitude toward housing a cat on their ward (*r* = 0.692, *p* = 0.006). Moreover, inpatients believed that the cat’s impact on ward atmosphere was higher if they saw the cat more often (*r* = 0.676, *p* = 0.002), and if they liked cats more in general (*r* = 0.486, *p* = 0.041). However, the inpatient’s emotional relationship with the cat did not correlate with how often they saw the cat (*r* = 0.205, *p* = 0.400). 

The inpatient’s pet history correlated with some items. Their previous experiences with pets correlated significantly with their perceived closeness to the cat (*r* = 0.567, *p* = 0.22) and their attitude toward housing a cat on their ward (*r* = 0.370, *p* = 0.48). However, having pets currently or in the past and growing up with animals did not correlate significantly with any ward cat-related item.

### 3.2. Inpatients Treated on Wards without Cats

The results of the open-ended questions revealed that all the inpatients were able to think of potential positive and potential negative aspects of housing a cat on their ward. Nine out of 14 inpatients thought of potential contact with a cat as positive (e.g., can be petted, you can play with the cat), and 7 inpatients believed that the cat might have a positive effect on their state of mind (e.g., cat has a calming effect, cat can be comforting). The most often mentioned negative aspects of potentially housing a cat on a psychiatric ward targeted allergies (4 out of 14 inpatients) or hygiene (4 out of 14 inpatients). 

The results of the rating questions showed that 8 out of 14 inpatients believed a cat would help them feel more comfortable in the psychiatric clinic, whereas 6 inpatients believed the cat would have a medium impact. Further, nine inpatients thought that a cat would positively influence their satisfaction with their ward. Four inpatients thought that the cat would have a medium effect, and two thought that the cat would have a rather negative effect on their satisfaction with the ward (see Table 3).

Analysis of the polar questions revealed that 12 out of 14 inpatients treated on wards without cats stated that they would prefer to have a cat on their own ward.

### 3.3. Staff Member on Wards with Cats

All the interviewed staff members working on wards with cats had positive feelings about the cat on their ward. Regarding the possible effects of the ward cat’s presence on inpatients, 16 of 16 staff members (one missing value) believed the cat helped inpatients feel more comfortable and led to more conversation, both between inpatients and between inpatients and staff members. Further, 14 out of 17 staff members thought the cat had a positive impact on inpatient emotional well-being and the ward atmosphere, whereas 3 believed the cat had a neutral impact.

Regarding the influence of the ward cat on staff members themselves, 14 out of 17 staff members stated the cat had a positive impact on their work satisfaction, 3 said that the cat had a neutral effect, and no one indicated a negative effect.

## 4. Discussion

The present study investigated inpatient and staff member attitudes toward and experiences with ward cats in a psychiatric setting to identify possible mechanisms on how cats affect patient satisfaction.

The interview results revealed that the inpatients treated and staff members working on wards with cats and inpatients treated on wards without cats all had positive attitudes toward ward cats. All the inpatients who were treated on a ward with a cat listed positive aspects of housing a cat, while only 6 out of 19 could think of potential negative aspects. The majority of inpatients liked having a cat on their ward, with only one patient preferring to live on a ward without a cat. However, not all the inpatients believed that the cat had a positive impact on their emotional well-being or on factors related to patient satisfaction [27,28,29,30,31], e.g., that the cat helps patients feel more comfortable or leads to more conversations between patients. There was also no difference in ward satisfaction between inpatients living on wards with cats compared to inpatients on wards without cats. These findings did not confirm the hypotheses found in the literature. Moreover, these results are in contrast to those of our previous study, where inpatients on wards with cats showed a greater overall patient satisfaction and were more satisfied with the common rooms, possibly indicating they perceived a better atmosphere [5]. This could relate to the small sample size or different measurement approach. Our previous study included data from more than 160 inpatients and measured patient satisfaction with a validated questionnaire, whereas this study only included 33 inpatients, and patient satisfaction was drawn from only one question. Since previous work also reported mixed results regarding the effects of animals on outcomes of mental health [25], it could be that a positive effect of cats on patient satisfaction depends highly on the individual patient.

Inpatient and staff member perception about the cat’s impact on ward atmosphere and on inpatient well-being were different. While patients did not perceive such an effect, staff members reported a positive influence of the cat on ward atmosphere and inpatient well-being. A possible explanation is that ward cats did not have a direct impact on inpatients, but worked as mediators by influencing staff members. Previous research reported beneficial effects of the presence of animals in health care centers [9,14,32]. Similarly, 14 of 17 staff members in the present study thought that the cat had a positive impact, while 3 thought that the cat had a neutral effect on work satisfaction. If the cat led to greater work satisfaction, it is possible that the staff member’s mood was better, which could in turn influence inpatients. This idea is supported in research documenting that nursing staff is more motivated and in a better mood when an animal is present [33]. Additionally, the presence of a companion animal can allow nursing staff to work more effectively, experience less stress, and spend more productive time with patients [34,35]. However, our data must be interpreted carefully, since staff members could be biased and over-interpret the effect of a cat. Staff members agreed with housing a pet on their ward and investing time to care for a cat. Therefore, it is likely that staff members had higher expectations regarding the effects of a cat.

Our results suggest that the ward cat’s impact on inpatients depends on additional factors. The correlational analysis showed that inpatients on wards with a cat perceived the cat’s impact on ward atmosphere to be higher when they reported a stronger emotional relationship with the cat or saw the cat more often. The ward atmosphere is related to patient satisfaction [28,29,30,31] and positively influences the number of coercive measures, aggressive incidents, unauthorized absences, therapeutic outcome, and compliance in patients [36,37,38].

Our finding indicates that a relationship with a cat, which is defined as extent of emotional closeness and how often the patient saw the cat, is a prerequisite that influences patient satisfaction. It could be that relationship with a cat provides a direct source of comfort [11], which is in line with studies describing better outcomes, e.g., an increase of plasma oxytocin from contact with familiar animals compared to unfamiliar animals [39,40]. Regular eye contact with a cat might also have an impact, since research suggests that the mere presence of an animal is sufficient to experience beneficial effects [41,42]. 

Surprisingly, the intensity of the relationship with the cat did not correlate with how often the patient saw the cat, illustrating that this study is not sufficient to clarify which factor is more important. Correlations must be interpreted carefully, as the number of interviews was small, and no objective measurements were used. 

This study intended to generate hypotheses for mechanisms linking ward cats and patient satisfaction. Our results showed that all the inpatients on wards with cats noticed the cats and perceived them as integral to the ward. The majority of inpatients on wards without cats preferred having a ward cat. This outcome builds on our previous results, showing that patient satisfaction is higher on wards with a cat compared to wards without a cat. 

Future research should prospectively investigate the effect of ward cats on patient satisfaction. It would be valuable to combine patient self-reporting with patient and cat behavioral measurements, such as how often, how long, and how they interact. Individual cat personality is likely an important factor, and future work should identify whether specific characteristics are important to inpatients and staff members, incorporating a One Health approach [43].

To make specific recommendations for clinics, more knowledge is needed about which types of inpatients might benefit from exposure to cats, and the extent of interaction and relationship necessary to improve patient satisfaction. Although we found inpatients were highly positive toward housing ward cats in general, this should always be addressed on an individual level. In our study, one inpatient on a ward with a cat preferred living on a ward without a cat, which should be respected regardless of the reported positive effects that animals might have on humans. Every healthcare center should have an animal-free ward, so patients can avoid contact when they are uncomfortable with animals.

## 5. Conclusions

Positive feedback from inpatients and staff members supports the idea that cats enhance patient satisfaction in psychiatric clinics. However, perceptions of the two groups differed regarding the effects and impact of ward cats. Our correlational analysis indicated that the impact of ward cats on patient satisfaction could depend on how frequently the patients see the cat, how close the relationship is with the cat, and whether patients like cats. Further research should investigate these hypotheses in a larger sample to make recommendations on integrating cats in psychiatric clinic treatment settings to enhance inpatient satisfaction. 

## Figures and Tables

**Table 1 ijerph-16-03108-t001:** Semi-structured interview with inpatients on wards with cats: Response distribution of the rating questions.

Item	No Approval/Negative/Not at All/Never (1–2)		Medium Approval/Neutral/Sometimes (3–4)		High Approval/Positive/Very Much/Very Often (5–6)	
	*N*	%	*N*	%	*N*	%
How often do you see the ward cat?	1	5.3	7	36.9	11	57.9
How close is your contact with the cat?	10	52.6	6	31.6	3	15.8
How do you feel about having a cat on your ward?	0	0	2	10.5	17	89.5
How does the ward cat affect your stay at the clinic?	0	0	13	68.4	6	31.6
How does the ward cat affect your emotional well-being?	0	0	10	52.7	9	47.4
Do you like cats?	1	5.3	3	15.8	15	79
How is your emotional relationship to the ward cat?	1	5.3	10	52.7	8	41.9
How much time do you spend daily with the ward cat?*	5	26.3	13	68.4	1	5.3
How would the ward atmosphere change if the cat was no longer there?	7	38.9	9	50%	2	11.1

*N* = number of inpatients; * not at all (1–2), around 10 minutes (3–4), around 60 minutes (5–6).

**Table 2 ijerph-16-03108-t002:** Semi-structured interview with inpatients on wards with cats: Response distributions of the polar questions.

Item	Yes		No	
	*N*	%	*N*	%
Does the ward cat help you to feel more comfortable in the clinic?	7	36.8	12	63.2
Does the ward cat lead to conversations between you and other inpatients?	8	42.1	11	57.9
Would you prefer if the ward cat was not living on your ward? *	1	5.3	17	89.5
Is the ward cat affectionate?	16	84.2	3	15.8
Do you search for the ward cat actively?	4	21.1	15	78.9
Would you prefer another ward animal?	4	21.1	15	79.8

*N* = number of inpatients; * One patient declined to answer the question because he felt that this would not be his decision.

**Table 3 ijerph-16-03108-t003:** Semi-structured interview with inpatients on wards without cats: Response distribution of the rating questions.

Item	No Approval/Negative/Not at All/Never (1–2)		Medium Approval/Neutral/Sometimes (3–4)		High Approval/Positive/Very Much/Very Often (5–6)	
	*N*	%	*N*	%	*N*	%
How do you think a ward cat could affect the ward atmosphere?	0	0	6	42.9	8	57.1
How do you think a ward cat could affect your satisfaction with your ward?	1	7.1	4	28.5	9	64.5
Do you think that you would feel more comfortable with a ward cat in the clinic?	4	28.5	2	14.2	17	57.1

*N* = number of inpatients.

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
