# Peer review of "“There Is a Cat on Our Ward”: Inpatient and Staff Member Attitudes toward and Experiences with Cats in a Psychiatric Ward"

_ijerph, 2019, doi:10.3390/ijerph16173108_

Round 1
Reviewer 1 Report
Introduction
The introduction needs much more development.
- What kind of patients are being referred to (line 30)?
- What kind of outcomes (line 31)?
- The authors should outline the potential mechanisms for animals improving clinical outcomes.
- There is a large literature on the effects of dogs and other animals on well-being in hospitals and nursing homes. That needs to be described in the introduction.
- As the researchers also survey the staff, the research on the effects of therapy animals on employees should also be described.
- The evidence for the benefits of animals to improve mental health outcomes is mixed, and this should be addressed.
Method
- If the questions are not going to be listed in a table within the manuscript, then there should at least be some description of what the questions were asking in addition to the type of question that it was (e.g., rating, polar, open). For example, how many questions asked about their attitudes toward housing the cat? How many questions for the inpatients’ beliefs about the effects of the cats?
- For the analysis, why were the answers dichotomized (e.g., negative, medium, high) instead of analyzing the values themselves? It has long been recommended that variables be kept as is for analysis rather than dichotomized (c.f., Macallum et al., 2002; Cohen, 1983; Royston, Altman, & Sauerbrei, 2006)
- Did the inpatients get to choose whether they were on the ward with the cat when they were admitted? This should be stated one way or the other because that could bias the results.
Results
- As a general stylistic note, if a number begins a sentence, it should be spelled out.
- Some aspects of Table 1 are confusing. For example, for the item, “How often do you see the ward cat?” the answers are categorized as No approval/negative/not at all to High approval/positive/very much which doesn’t make sense as answers to that question.
- In Table 2, “Is the ward cat confidently?” – I don’t know what this means.
- For inpatients treated on wards without cats, did any inpatients report negative feelings about having a cat on their ward or that a cat would decrease their satisfaction? The whole picture was provided for the inpatients with cats on their ward so a similar amount of information should be presented for the other group. Likewise this data could be added as another column in the tables.
- A large section of the survey dealt with the pet history of the participants but this is not mentioned in the results section? Were analyses conducted to assess whether pet history was correlated with opinions about ward cats? This should be addressed.
- What about the analysis of the open-ended questions? What were the results?
- Again, I think that the analyses should be conducted without the dichotomization of the rating scales.
- Was there a comparison of overall ward satisfaction between the two wards? E.g., were people on the ward with the cat more satisfied than those on the ward without the cat? This seems important to the aims of the paper.
Discussion
- I think it’s a leap to say that the patient satisfaction is “unconscious” because other studies did find patient effects.
- There is literature showing the positive effects of therapy animals on employees, this could be addressed here.
- Lines 150-152 “The results of the interviews revealed that inpatients treated and staff members working on wards with cats, as well as inpatients treated on wards without cats, all had positive attitudes toward ward cats.” Is this accurate? Because in Table 1 it shows that one patient had a negative relationship with the ward cat. This needs to be revised as not everyone was happy with the ward cat.
- Because there was a patient that was not happy with the ward cat, how should healthcare centers address these issues? This is an important topic to cover.
Author Response
Major revision
Dear reviewer
We are pleased to resubmit manuscript after major revision. We thank you for your helpful comments that clearly improved our manuscript. We addressed all the comments as listed below and made the changes in color within our manuscript.
Introduction
The introduction needs much more development.
What kind of patients are being referred to (line 30)?
This refers to inpatients with mental illness. We put this information into the introduction (line 31).
What kind of outcomes (line 31)?
We added examples based on previous research in line 32-33.
The authors should outline the potential mechanisms for animals improving clinical outcomes.
Thank you for this suggestion. We extended this point in the introduction and added research addressing this topic (see lines 36-43).
There is a large literature on the effects of dogs and other animals on well-being in hospitals and nursing homes. That needs to be described in the introduction.
Thank you for this suggestion. We extended the information about the effects of dogs and other animals on well-being in hospital and nursing homes (line 39-41).
As the researchers also survey the staff, the research on the effects of therapy animals on employees should also be described.
Thank you for this comment. We briefly commented it in the introduction (line 42-43) and included previous research on the effects of therapy animals on employees in our discussion where we focus on this aspect (line 234-246).
The evidence for the benefits of animals to improve mental health outcomes is mixed, and this should be addressed.
Thank you for this comment. We addressed the mixed evidence shortly in the introduction (see line 42) and extended it in the discussion (see line 228-230).
Method
If the questions are not going to be listed in a table within the manuscript, then there should at least be some description of what the questions were asking in addition to the type of question that it was (e.g., rating, polar, open). For example, how many questions asked about their attitudes toward housing the cat? How many questions for the inpatients’ beliefs about the effects of the cats?
Thank you for this suggestion. We included this information into the manuscript in line 73-107.
For the analysis, why were the answers dichotomized (e.g., negative, medium, high) instead of analyzing the values themselves? It has long been recommended that variables be kept as is for analysis rather than dichotomized (c.f., Macallum et al., 2002; Cohen, 1983; Royston, Altman, & Sauerbrei, 2006)
We absolutely agree. We did not use dichotomization for the statistical analyses but rather used a dichotomization for interpreting and reporting the results in an easier way. This was stated misleading and we changed it (see line 108-116). Thank you very much for this helpful comment.
Did the inpatients get to choose whether they were on the ward with the cat when they were admitted? This should be stated one way or the other because that could bias the results.
Thank you for this very important comment. No, inpatients were not able to choose if they wanted to stay on a ward with or without cat. We previously had stated this in the results section but relocated this information into the methods section to line 58-60.
Results
As a general stylistic note, if a number begins a sentence, it should be spelled out.
We corrected this and did spell out every number that begins a sentence.
Some aspects of Table 1 are confusing. For example, for the item, “How often do you see the ward cat?” the answers are categorized as No approval/negative/not at all to High approval/positive/very much which doesn’t make sense as answers to that question.
Thank you for this helpful comment. We added the answer categories “never”, “sometimes” and “very often”.
In Table 2, “Is the ward cat confidently?” – I don’t know what this means.
Thank you for this feedback. We changed the translation of the word “zutraulich” in German into “affectionate".
For inpatients treated on wards without cats, did any inpatients report negative feelings about having a cat on their ward or that a cat would decrease their satisfaction? The whole picture was provided for the inpatients with cats on their ward so a similar amount of information should be presented for the other group. Likewise this data could be added as another column in the tables.
Thank you for this feedback. Yes, even though most of the interviewed inpatients would like to have a cat on their own ward they were able to think of hypothetical negative aspects regarding housing a cat on their own ward, e.g. allergies, the cat’s welfare or if the cat brings insects to the ward (182-188). We extended the results for inpatients on wards without cats and also added this data in another table (see Line 189-199).
A large section of the survey dealt with the pet history of the participants but this is not mentioned in the results section? Were analyses conducted to assess whether pet history was correlated with opinions about ward cats? This should be addressed.
Yes, analyses were conducted. Based on your suggestion we included pet history in our result section (line 177-180).
What about the analysis of the open-ended questions? What were the results?
Not all open-ended questions were cat-related. Therefore, we only included the two open questions thatasked inpatients and staff members about positive and negative aspects of housing a cat on their ward in the manuscript.
Again, I think that the analyses should be conducted without the dichotomization of the rating scales.
We did not use dichotomization for the analyses e.g. the correlations. However, we used a dichotomization for interpreting the results. This was stated very misleading in the methods section. We corrected it and thank you for this important comment.
Was there a comparison of overall ward satisfaction between the two wards? E.g., were people on the ward with the cat more satisfied than those on the ward without the cat? This seems important to the aims of the paper.
Thank you for this suggestion. Results of our semi-structured interview showed no statistically significant differences in satisfaction between patients on wards with and without a cat. We added this information into the result section (line 132-134) and included it in the discussion (line 222-230).
Discussion
I think it’s a leap to say that the patient satisfaction is “unconscious” because other studies did find patient effects.
We deleted this point in our discussion and replaced it with other possible explanations (Line 234-248).
There is literature showing the positive effects of therapy animals on employees, this could be addressed here.
We included recent research on that topic in the discussion (line 234-248).
Lines 150-152 “The results of the interviews revealed that inpatients treated and staff members working on wards with cats, as well as inpatients treated on wards without cats, all had positive attitudes toward ward cats.” Is this accurate? Because in Table 1 it shows that one patient had a negative relationship with the ward cat. This needs to be revised as not everyone was happy with the ward cat.
Thank you, we changed this and added “the majority of…” (line 218).
Because there was a patient that was not happy with the ward cat, how should healthcare centers address these issues? This is an important topic to cover.
This is a highly important topic for practice, thank you for pointing this aspect out. We addressed this in line 277-281.
Reviewer 2 Report
There is a tonne of research already in this area regarding the psychosocial benefits of therapy animals, and some of this previous data could be used to enrich the introduction
The language used in this article, particularly in the discussion and conclusion, is overly complex and inaccessible, and should be simplified for readability and comprehension
Author Response
Dear reviewer
We are pleased to resubmit manuscript after major revision. We thank you for your helpful comments that clearly improved our manuscript. We addressed all the comments as listed below and made the changes in color within our manuscript.
Introduction:
There is a tonne of research already in this area regarding the psychosocial benefits of therapy animals, and some of this previous data could be used to enrich the introduction.
We extended the inclusion of previous research in the introduction as well as the discussion, see line 36-42 and 238-245; line 256-260
Discussion and Conclusion:
The language used in this article, particularly in the discussion and conclusion, is overly complex and inaccessible, and should be simplified for readability and comprehension.
We did have the manuscript checked and corrected again by a native English-speaking scientist.
Reviewer 3 Report
The welfare of the cats need to be addressed at least briefly. One aspect is how the cats are kept, they are shut in at night, where how long, etc etc, during day are they outside or in wards, are they fed in the wards etc etc.
All the interviews are verbal, I would suggest that since many people, particularly those mentally disturbed lie about their experiences etc, that it is important to look more carefully at the behaviour of the inmates and the nurses to the cats, how long touching, finding, looking at, etc etc, and how the cat responds :purring or not, moving away, moving towards, etc does this matter to the inmates etc etc.
The results are not unexpected and with some additions ( as above) this paper should be published, but to go forward and make recommendations it needs to branch out to measure behaviours of the cats, inmates and nurses to the cat and the cats responses... not all cats behave the same way, this paper seems to have lost sight of the fact that the cat is a living moving individual being, and not a commuter who might act in the same way always. This MUST be addressed before publication.
Author Response
Dear reviewer
We are pleased to resubmit manuscript after major revision. We thank you for your helpful comments that clearly improved our manuscript. We addressed all the comments as listed below and made the changes in color within our manuscript.
Comments and Suggestions for Authors
The welfare of the cats need to be addressed at least briefly. One aspect is how the cats are kept, they are shut in at night, where how long, etc etc, during day are they outside or in wards, are they fed in the wards etc.
Thank you for your feedback. We agree, the welfare of cat should be addressed especially because the animal’s welfare is such an important factor. We extended the information about that (see line 60-64). Since every cat is different we do not know how long the cats would stay outside and where they went.
All the interviews are verbal, I would suggest that since many people, particularly those mentally disturbed lie about their experiences etc, that it is important to look more carefully at the behaviour of the inmates and the nurses to the cats, how long touching, finding, looking at, etc etc, and how the cat responds :purring or not, moving away, moving towards, etc does this matter to the inmates etc etc.
Thank you for sharing this interesting thought. Indeed, there is a chance that mentally disturbed people might lie about their experiences. In this study, we can only rely on the patients’ verbal answers. This was why we wanted to also have an outside view by interviewing staff members. Combining patients’ self-report with actual behavioral measures could be implemented in future studies, we have added this in line 271-277.
We did not ask patients about how the cat responded to their behavior. We only asked them if the cat was affectionate since we wanted to know if there was at least the possibility for inpatients and staff members to interact with the cat. It would however been interesting to detect the cat’s behavior in a more detailed manner and should be a topic for further investigations. We have added this suggestion in line 271-277.
The results are not unexpected and with some additions ( as above) this paper should be published, but to go forward and make recommendations it needs to branch out to measure behaviours of the cats, inmates and nurses to the cat and the cats responses... not all cats behave the same way, this paper seems to have lost sight of the fact that the cat is a living moving individual being, and not a commuter who might act in the same way always. This MUST be addressed before publication.
Thank you for this feedback and your suggestion. It was definitively not our intention to elicit this impression. We extended our suggestions for future work in line 271-277 regarding this aspect and proposed to implement a clear One Health focus as well as closely look into specific characteristics of the interaction and its dependency from the individuals.
Round 2
Reviewer 3 Report
My comments have been attended to, if rather briefly. I still have the reservation concerning no cat behaviour was recorded, let us hope that they take this further next time, but yes Ok to publish now.